# Cyclophilin A Is Not Acetylated at Lysine-82 and Lysine-125 in Resting and Stimulated Platelets

**DOI:** 10.3390/ijms23031469

**Published:** 2022-01-27

**Authors:** Annabelle Rosa, Elke Butt, Christopher P. Hopper, Stefan Loroch, Markus Bender, Harald Schulze, Albert Sickmann, Sandra Vorlova, Peter Seizer, David Heinzmann, Alma Zernecke

**Affiliations:** 1Institute of Experimental Biomedicine, University Hospital Würzburg, 97080 Würzburg, Germany; rosa_a@ukw.de (A.R.); butt_e@ukw.de (E.B.); christopher.p.hopper@gmail.com (C.P.H.); Bender_M1@ukw.de (M.B.); harald.schulze@uni-wuerzburg.de (H.S.); vorlova_s@ukw.de (S.V.); 2Leibniz-Institut für Analytische Wissenschaften (ISAS), 44139 Dortmund, Germany; stefan.loroch@isas.de (S.L.); albert.sickmann@isas.de (A.S.); 3Medizinisches Proteom-Center, Ruhr-University Bochum, 44801 Bochum, Germany; 4Department of Chemistry, College of Physical Sciences, University of Aberdeen, Aberdeen AB24 3FX, UK; 5Hospital Ostalb gkAöR, 73430 Aalen, Germany; Peter.Seizer@kliniken-ostalb.de; 6Department of Cardiology and Angiology, University of Tübingen, 72076 Tübingen, Germany; David.Heinzmann@med.uni-tuebingen.de

**Keywords:** Cyclophilin A, acetylation, platelets, CD147, EMMPRIN

## Abstract

Cyclophilin A (CyPA) is widely expressed by all prokaryotic and eukaryotic cells. Upon activation, CyPA can be released into the extracellular space to engage in a variety of functions, such as interaction with the CD147 receptor, that contribute to the pathogenesis of cardiovascular diseases. CyPA was recently found to undergo acetylation at K82 and K125, two lysine residues conserved in most species, and these modifications are required for secretion of CyPA in response to cell activation in vascular smooth muscle cells. Herein we addressed whether acetylation at these sites is also required for the release of CyPA from platelets based on the potential for local delivery of CyPA that may exacerbate cardiovascular disease events. Western blot analyses confirmed the presence of CyPA in human and mouse platelets. Thrombin stimulation resulted in CyPA release from platelets; however, no acetylation was observed—neither in cell lysates nor in supernatants of both untreated and activated platelets, nor after immunoprecipitation of CyPA from platelets. Shotgun proteomics detected two CyPA peptide precursors in the recombinant protein, acetylated at K28, but again, no acetylation was found in CyPA derived from resting or stimulated platelets. Our findings suggest that acetylation of CyPA is not a major protein modification in platelets and that CyPA acetylation is not required for its secretion from platelets.

## 1. Introduction

Cyclophilin A (CyPA) is the best-known member of the cyclophilin family and is expressed by all known prokaryotic and eukaryotic cells, accounting for ~0.4% of the total cytosolic protein fraction [1]. CyPA was initially identified as a binding protein of the immunosuppressive drug Cyclosporin [2]. Intracellular CyPA possesses peptidyl-prolyl cis-trans isomerase (PPIase) activity and is involved in functions such as protein folding and trafficking, which collectively contribute to various cellular processes including cytokinesis, T-cell subtype differentiation, and platelet activation. Upon inflammatory stimulation, however, CyPA can also be secreted, e.g., by endothelial cells (ECs) and vascular smooth muscle cells (VSMCs) to subsequently engage in extracellular functions, such as chemotactic effects on leukocytes [3,4].

Extracellular CyPA has been involved in the pathophysiology of different cardiovascular diseases, including cardiovascular remodeling after myocardial ischemia, cardiac hypertrophy, abdominal aortic aneurysm formation, atherosclerosis, and vascular remodeling [5]. These effects are mediated by activation of CD147 on target cells, which is also known as an extracellular matrix-metalloproteinase inducer (EMMPRIN) [6,7].

Platelets are anucleated cells derived from both bone marrow and lung megakaryocytes [8] with a variety of physiological activities, most notably essential roles in hemostatic plug formation to avoid excessive blood loss after vascular injury. Platelet thrombus formation involves the activation, aggregation, and adherence of platelets to the forming thrombus. Platelets can, however, also cause thrombotic occlusion or obstruction of the blood flow in arteries via activation after atherosclerotic plaque erosion or rupture, which is an underlying mechanism of myocardial infarction and stroke [9]. Beyond hemostasis and thrombosis, platelets play essential roles in multiple inflammation-related diseases and contribute to atherosclerosis, myocarditis, and ischemia–reperfusion injury. For instance, platelets can release inflammatory mediators, such as cytokines and damage-associated molecular patterns (DAMPs), upon activation to recruit immune cells and modulate inflammation at lesion sites [10,11].

Within platelets, intracellular CyPA functions as a mediator of thrombus formation by regulating Ca^2+^ mobilization from intracellular stores [12]. In addition, CyPA controls integrin αIIbβ3 bidirectional signaling and cytoskeletal remodeling [13]. Upon activation, CyPA can also be released from platelets [14]. Notably, higher CyPA plasma concentrations were measured in patients with acute stroke [15,16,17,18,19,20,21,22,23] (as summarized in Appendix A), and platelet surface expression of CyPA has been found to be positively correlated with hypertension, hypercholesterolemia, and increased mortality of patients with symptomatic coronary artery disease (CAD) [24].

Proteins frequently undergo post-translational modifications. Acetylation is found in 90% of higher eukaryotic proteins [25]. In isolated human platelets, more than 270 acetylated proteins that are involved in platelet actin cytoskeletal remodeling, and metabolic processes were identified by proteomics, which included acetylated CyPA with acetylation at K28 and K82. Lysine acetylation is catalyzed by lysine acetyltransferases (KATs) [25]. Acetyltransferase KAT/p300 was found to play a major physiological role for lysine acetyltransferase activity in platelet function [26].

In VSMC cells, CyPA was found to be acetylated upon angiotensin II (AngII) stimulation. Three out of five critical lysine residues of CyPA are conserved in most species [3]. Among these, site-directed mutagenesis and lentiviral transduction of VSMCs with respective mutants identified K82 and K125 on the surface of the protein as the predominant CyPA residue acetylated in response to Ang II, and to be required for CyPA secretion in response to oxidative stress induced by Ang II stimulation [22]. Soe et al. further showed that CyPA secreted from VSMCs transduced with K82/125R mutants, as well as in vitro acetylated recombinant CyPA, showed greater activation of VSMCs than non-acetylated CyPA [22]. Subsequently, acetylated CyPA was also shown to be a more potent agonist for endothelial cells [20]. As activated platelets are known to secrete CyPA for local delivery of CyPA during cardiovascular disease events [7], and given the pronounced inflammatory activity of extracellular CyPA observed under experimental conditions, we hereby addressed whether acetylation is also important for CyPA release from platelets.

## 2. Results

### 2.1. Western Blot Analyses of CyPA Acetylation in Platelets

It has been shown previously that more than 270 proteins involved in actin cytoskeletal remodeling are acetylated in unstimulated platelets [26]. To study the acetylation of CyPA in activated platelets, we stimulated human and mouse platelets with 0.5 U/mL thrombin and analyzed CyPA acetylation by Western blotting. Stimulation of washed human and mouse platelets with thrombin resulted in platelet activation, as confirmed by increased amounts of phosphorylated p38 in the cell lysate. CyPA was present in equal amounts in both untreated and thrombin-stimulated platelet lysates, and a very weak band of acetylated protein at the expected size of CyPA at 16 kDa could be detected (Figure 1). Another band of acetylated protein was visible at around 14 kDa (Figure 1). It should be noted that basal lysate acetylation was donor dependent and seen only in two out of six experiments.

Owing to the low affinity of acetyllysine (acK) antibodies for varying acK motifs and the high abundance of some acetylated proteins [27], we included an immunoprecipitation (IP) step for CyPA in the run-up. Cell lysates were subjected to IP with monoclonal CyPA antibody and analyzed by Western blotting using polyclonal CyPA antibody, polyclonal acK antibody, and protein A-HRP as a secondary antibody, to avoid cross-reactivity with IP heavy and light chains. In immunoprecipitated lysates, CyPA was readily detected in human platelets. However, no acetylation at 16 kDa corresponding to CyPA was detected. This suggests that other proteins with similar molecular weights might have been detected when evaluating acetylation in the lysate (Figure 1). A similar observation was reported by Soe et al. when investigating VSMCs [22]. We also noted an acetylated band at around 14 kDa.

Unexpectedly, we could not immunoprecipitate CyPA from mouse platelet lysates (Figure 1), despite 96.5% protein sequence homology between humans and mice and the application of three different primary antibodies against CyPA.

### 2.2. Recombinant CyPA Acetylation by p300

In platelets, protein lysine acetylation is mediated by the lysine acetyltransferase p300 [26]. We performed in vitro acetylation experiments with recombinant human CyPA and the catalytic subunit p300 acetyltransferase in the presence of acetyl-CoA. Western blot analyses demonstrated high auto-acetylation of p300, but CyPA acetylation was undetectable at the expected size of 16 kDa (Figure 2). A lower band of acetylated protein was detected at around 14 kDa in the presence of both p300 and acetyl-CoA, but this band was not detected by the total CyPA antibody.

### 2.3. Mass Spectrometric Analyses of CyPA

To overcome the limitation of not being able to detect CyPA in mouse IP platelet lysates, we capitalized on the high expression of CyPA in murine platelets and separated the mouse CyPA on a 13% polyacrylamide gel, followed by mass spectrometry-compatible silver staining and Western blotting, to detect single bands around 16 kDa. In parallel, for reference, a human lysate probe was loaded on the gel and processed simultaneously for Western blotting and silver staining (Figure 3A).

Using shotgun proteomics in conjunction with a data-dependent acquisition, we detected two CyPA peptide precursors with lysine acetylation when analyzing in vitro acetylated recombinant human CyPA. The precursor [VSFELFADK(ac)VPK+2H]2+, describing acetylation of K28, was the only promising candidate identified with two spectra, a mass deviation of +0.81 and +0.41 ppm and Mascot ion scores of 59 and 41 (Figure 3B). The other precursor [Vk(ac)EGM(ox)NIVEAM(ox)ER+2H]2+, describing acetylation of K133, was quite likely to be false positive: (I) It was identified with a mass deviation of −0.07 ppm and −1.95 ppm (average mass deviation in the run was +1.2 ppm) (Figure 3C), (II) exhibited a non-matching precursor isotope pattern, and (III) exhibited low Mascot ion scores of 35 and 22. In contrast, neither human- nor mouse-derived stimulated platelet samples yielded any lysine-acetylated peptide identification. For higher specificity and sensitivity of detection, we performed targeted LC–MS in parallel reaction monitoring-mode targeting acetylation on K28, K82, and K125. We triggered the doubly and triply charged ion of the peptides VSFELFADK(ac)VPK), SIYGEK(ac)FEDENFILK, and TEWLDGK(ac)HVVFGK (Figure 3D, highlighted in blue). We analyzed in vitro acetylated CyPA and stimulated platelet samples (mouse and human) after IP for the enrichment of CyPA. Among all samples, K28 was exclusively detected in the in vitro acetylated sample but in none of the stimulated platelet samples. Moreover, in none of the samples could we detect any appreciable ion traces for K82 or K125 acetylation, which is in good accordance with the results from our data-dependent acquisition approach.

### 2.4. CyPA Release

Finally, we analyzed CyPA secretion from thrombin-activated human platelets. After stimulation, the released proteins were separated from platelets by centrifugation. The purity of the fraction was confirmed by the absence of platelet membrane-specific protein GPαIIb (Figure 4). Western blot analysis demonstrated the presence of CyPA in resting and activated platelet pellets, as well as in the supernatants of activated platelets. However, no acetylation was detected in the pellet lysate or the supernatant (Figure 4).

The data are in stark contrast to other vascular cell types, e.g., VSMCs studied by Soe et al. that showed an acetylation-dependent release of CyPA into the medium [22]. Stimulation of human aortic smooth muscle cells (HASMCs) with 300 nM Ang II resulted in a modestly increased ERK phosphorylation after 2 h of stimulation but reduced levels after 24 h. Simultaneously, we observed acetylation of CyPA in cell lysates and in immunoprecipitated CyPA from cell lysates, and a release of CyPA into the medium (Figure 5).

## 3. Discussion

Several papers describe acetylation of CyPA in human cells and platelets [22,26,28,29]. Herein we assessed the acetylation of CyPA in platelets and its requirement for its release from activated platelets. By Western blotting, we observed a faint band of an acetylated 16 kDa protein in lysates of control and thrombin-stimulated human platelets that corresponded in size to CyPA Western blot bands. This low acetylation was inconsistently detected and could potentially be in agreement with mass spectrometry data detecting basal acetylation of CyPA at K28 and K82 [26]. Acetylation of K28 was also seen in recombinant CyPA after p300 stimulation in our analyses. However, we did not observe acetylation of CyPA at K28, K82, or K125 in resting platelets or after stimulation.

An acetylated band around 14 kDa was also seen sporadically in human platelet lysates, and after incubation with p300. However, no CyPA band was detected at the same size as a confirmation in platelet lysates. Interestingly, very low amounts of N-terminally truncated form of the protein (8–14 kDa) were identified in cultured mouse glial cells, which, in contrast to full-length CyPA (16 kDa), induces a distinct pattern of cytokine release in cultured microglia or astroglia [30]. The acetylated 14 kDa protein observed in our experiments could thus be in line with a truncated form of CyPA. However, three different monoclonal and polyclonal antibodies against CyPA detected the 16 kDa but not the 14 kDa form in human and mouse platelet lysates, so it is unlikely that CyPA is the protein that was acetylated at this size.

Upon platelet activation, platelets rapidly release their granule content. Alpha granules release their contents into the open canicular system (OCS) at the platelet surface, while dense granules fuse directly with the membrane to exocytose activation mediators, e.g., coagulation proteins, cytokines, and proinflammatory molecules [31]. CyPA was found in α-granules by mass spectrometry [32], and we could observe CyPA secretion from platelets as early as after 5 min of thrombin stimulation. In addition, CyPA regulates Ca^2+^ mobilization from intracellular stores [12], integrin αIIbβ3 bidirectional signaling, and cytoskeletal remodeling [13], in line with intracellular and cytosolic functions of CyPA.

Quantification of acetylated versus non-acetylated CyPA in Jurkat and Hela cells revealed only 38–50% of intracellular CyPA to be acetylated [28]. Lysine acetylation patterns in different subcellular locations vary in between tissues and organs; acetylated CyPA was exclusively found to be cytoplasmic in different tissues, based on the subcellular localization as defined by the GO term cellular component [33]. This may suggest that CyPA is acetylated intra-cytoplasmically in platelets rather than within α-granules and that only non-acetylated granular CyPA is released from activated platelets, in line with the absence of acetylated CyPA in platelet supernatants. However, we cannot fully exclude rapid deacetylation in the supernatant, and NAD-dependent deacetylases are present in serum [34].

Acetylsalicylic acid (ASA) is known to irreversibly acetylate COX in platelets at serine residues within the active site. ASA has also been described to acetylate a lysine in CyPA in human colon cancer cell lines, so that in principle ASA may also acetylate CyPA in platelets [35,36,37]. However, preincubation of platelets with ASA results in a broad decrease in proteins released by activated platelets, and no CyPA can be detected in the supernatant of ASA-treated platelets [38].

In VSMCs, acetylated CyPA can be found after 4 h of Ang II treatment, and release of CyPA is noted after 16 h of stimulation. In our control experiments, we also detected CyPA acetylation after Ang II treatment in HASMCs and observed secretion of CyPA into the cell supernatant (Figure 5). CyPA is secreted via a highly regulated pathway that involves vesicle transport and plasma membrane binding, dependent on actin remodeling, myosin II activation, and vesicle-associated membrane protein 2 [39]. By overexpressing mutated CyPA carrying arginine instead of lysine at K82 and K125, acetylation at these positions was found to be required for the secretion of acetylated CyPA [22]. Quantification of the amount of acetylated versus non-acetylated CyPA, however, was not performed, so it is unclear to what extent these variants are found among secreted CyPA. The slow time dynamics of the release, however, argue for a mechanism of secretion different from the rapid release from platelets. Nevertheless, it would be interesting to assess CyPA acetylation in platelets in chronic diseases such as hypertension in the future, to assess potential long-term modifications that could alter intracellular CyPA acetylation and function in platelets. The rapid activation of platelets in cell culture, however, precludes prolonged times of incubation, as applied with VSMCs by Soe et al. [22].

Our data suggest an acetylation-independent release of the protein from activated platelets. Neither increased CyPA acetylation in supernatants of platelets nor in the platelet lysate was found after stimulation. Our findings thus describe low levels of acetylation of CyPA in platelets and, in contrast to VSMC, no CyPA acetylation at K82 or K125, indicating that CyPA acetylation at these sites is not prerequisite for its rapid secretion from platelets.

## 4. Materials and Methods

### 4.1. Platelet Preparation

Anonymized residual waste of citrated blood was obtained from healthy donors, participating in a study performed at the University of Würzburg (Ethics Committee Approval Number 52/15). At no time point, any backtracking to the donors (name, sex, age) was possible. Blood was centrifuged at 330× *g* for 15 min to obtain platelet-rich plasma, and platelets were subsequently collected by centrifugation at 400× *g* for 7 min and resuspended in Tyrode’s buffer (10 mM Hepes pH 7.4, 137 mM NaCl, 2.7 mM KCl, 5 mM glucose, 1 mM EDTA), at a density of 1 × 10^9^ cells/mL.

To obtain a sufficient number of platelets for protein analyses, blood from 4 C57BL/6 mice (8–12 weeks old), randomly selected from different cages and healthy by visual inspection, was collected. Mice were housed within the Zentrum für Experimentelle Molekulare Medizin (ZEMM) at the University of Würzburg, in standard cages containing hardwood chip bedding and enriched with mouse homes and cellulose tissue. Mice had ad libitum access to water and standard chow. Mice were maintained on a 12 h (hr) light/12 h dark cycle, at a temperature of 20–24 °C with 45–62% humidity. Mice that had not been subject to any sort of treatment and were sacrificed for scientific purposes did not require ethical approval by local authorities. Mice were euthanized by an overdose of isoflurane anesthesia (5% concentration) until one minute after breathing stops, and subsequent exsanguination by the punctuation of the heart with a syringe, preloaded with 100 µL citrate buffer. Pooled blood was diluted 1:1 with Tyrode’s buffer and centrifuged at 300× *g* for 10 min to obtain platelet-rich plasma. Platelets were subsequently collected by centrifugation at 720× *g* for 6 min and resuspended in Tyrode’s buffer at a density of 1× 10^9^ cells/mL.

### 4.2. Immunoprecipitation

Washed platelets (5 × 108 cells/mL) were stimulated with 0.5 U/mL thrombin for 5 min. Cells were harvested and lysed in 500 µL in IP buffer (20 mM Tris pH 7.4; 120 mM NaCl, 1% Triton, 5 mM EDTA) substituted with Complete Mini ((Roche, Mannheim, Germany)) and 10 µM EX527 acetylase inhibitor (Selleck Chemicals, Houston, TX ,USA), known to increase CyPA acetylation in HeLa cells [40]. Cell lysate (500 µL) was subjected to immunoprecipitation with a 2 µg monoclonal CyPA antibody (ab58155, Abcam, Cambridge, UK) for 2 h at 4 °C, followed by 2 h incubation with protein A/G sepharose beads (sc-2003, Santa Cruz, Dallas, TX, USA). Washed beads were stored at −80 °C for subsequent mass spectrometry analysis. For Western blot analysis, beads were resolved in 50 µL Laemmli buffer (10× concentration). For detection of CyPA release into the supernatant of thrombin-stimulated platelets, a cell density of 8 × 10^8^ platelets/mL was used. Following activation, platelets were removed by 1000× *g* centrifugation. The purity of the supernatant was confirmed by the absence of platelet membrane-specific GPαIIb protein.

### 4.3. In Vitro Acetylation Assay

Reactions (20 µL) containing 50 nM recombinant human CyPA (3589-CA, R & D Systems, Minneapolis, MN, USA), 1.2 mM acetyl-CoA (A2056, Sigma Aldrich, St. Louis, MO, USA), and 1 µg p300 catalytic domain (BML-SE451, Enzo Life Science, Lörrach, Germany) in HAT buffer (50 mM Tris pH7.5, 150 mM NaCl. 1 mM PMSF, 1 mM DTT, 10 mM sodium butyrate and 10% glycerol) were incubated at 30 °C for 30 min.

### 4.4. Western Blot

The cell lysate was resolved by SDS polyacrylamide gel electrophoreses. Proteins were transferred onto nitrocellulose and blocked subsequently with 5% milk in Tris-buffered saline/0.1% Tween20 (TBS/T) for 1 h. Next, blots were incubated overnight at 4 °C with appropriate antibodies: rabbit polyclonal acK antibody (#9441, Cell Signaling, Frankfurt am Main, Germany); rabbit polyclonal CyPA (BML-SA296, EnzoLife Science, Lörrach, Germany); p-P38 (#9211, Cell Signaling, Frankfurt am Main, Germany); β-Actin (#1616, Santa Cruz, Dallas, TX, USA); GPαIIb (kind gift of B. Nieswandt, University of Würzburg, Germany).

After washing 3 times with TBS/T, membranes were probed with secondary horseradish peroxidase-conjugated goat anti-rabbit or goat anti-mouse antibody (Biorad, Feldkirchen, Germany), and rabbit HRP protein A (PA1-26853, Thermo Fischer Scientific, Waltham, MA, USA), and visualized using enhanced chemiluminescence reagent (GE Healthcare, München, Germany). Chemiluminescence images were taken using the Amersham Imager 600 (GE Healthcare, München, Germany). Uncropped Western Blot images are provided as Appendix A.

### 4.5. Cultivation of Primary Aortic Smooth Muscle Cells (HASMCs)

Human primary aortic smooth muscle cells (HASMCs, ATCC PCS-100-012) were purchased from American Type Culture Collection (Manassas, VA, USA). Cells were cultured in vascular basal media (ATCC PCS-100-030) supplemented with vascular smooth muscle growth kit (ATCC PCS-100-042), containing 10% FCS, 10 U/mL penicillin, and 10 μg/mL streptomycin, at 37 °C under 5% CO_2_. HASMCs were used between passages 3 and 6 and growth arrested with Dulbecco’s modified Eagle medium/nutrient mixture F-12 (DMEM/F-12) containing 0.3% FBS overnight before the medium was changed to serum-free DMEM/F-12 one hour before the experiment. Cells were stimulated with Ang II (300 nM) for 2 and 24 h. The supernatant was collected, complemented with 10 µM EX527 and processed for immunoprecipitation similar to platelet supernatants. Cell lysates were washed with PBS and subjected to immunoprecipitation and Western blot as described for platelets.

### 4.6. Silver Staining

Silver staining was performed using the PierceTM Silver Stain for Mass Spectrometry Kit (#24600, Thermo Fischer Scientific, Waltham, MA, USA) according to the manufacturer‘s instructions. Gels were illuminated by a lightbox, and protein bands were excised with a clean scalpel and placed in 0.5 mL microcentrifuge tubes for in-gel trypsin digestion and mass spectrometry analysis.

### 4.7. Processing of Gel Bands for LC–MS

Cutout gel bands were washed alternately 3× with 50 mM ammonium bicarbonate (ABC), pH 7.8, and 25 mM ABC, pH 7.8/50% acetonitrile (ACN). Gel bands were dried, digested for 14 h with 125 ng Trypsin Gold (Promega, Walldorf, Germany)) in 50 mM ABC, and subjected to nanoLC–MS/MS. For peptide recovery, gel bands were incubated twice in 20 µL 0.1% TFA at 37 °C for 15 min followed by 25 µL 0.1% TFA/60% ACN. The resulting peptide fractions were combined, dried, and redissolved in 0.1% TFA for LC–MS.

### 4.8. Data-Dependent Acquisition

Liquid chromatography–mass spectrometry (LC–MS) was conducted using a U3000 RSLCnano ProFlow system online-coupled to an LTQ Orbitrap Velos Pro for data-dependent acquisition (DDA) or to a Q Exactive HF mass spectrometer (both Thermo Scientific, Bremen, Germany, including HPLC columns) for parallel reaction monitoring (PRM). For DDA, samples were loaded in 0.1% TFA at a flow rate of 30 µL/min. After 5 min, the pre-column was switched in line with the main column (Acclaim PepMap100 C18; 75 μm × 50 cm), and peptides were separated using a 90 min binary acetonitrile gradient ranging from 2% to 32% ACN in presence of 0.1% formic acid at 60 °C and a flow rate of 250 nL/min. The LTQ Orbitrap Velos Pro was operated in data-dependent acquisition mode, starting with a survey scan with a resolution of 60,000 (@ *m*/*z* 400), followed by up to 10 low-resolution MS/MS of the most intense precursor ions (top N) using collision-induced dissociation. AGC target values were set to 106 and 105 for MS and MS/MS, respectively, and the maximum injection time was set to 100 ms. Singly charged ions and ions of unassigned charge were excluded from fragmentation, and fragmented precursors were excluded from refragmentation for 20 s (dynamic exclusion). The lock mass at *m*/*z* = 371.1012 was used as an internal calibrant.

For targeted LC–MS, we used a 45 min binary gradient and a Q Exactive HF operated in parallel reaction monitoring mode (PRM). The resolution was set to 120,000 and 60,000 (at 200 *m*/*z*), with maximum ion injection times of 128 and 256 ms for MS and MS/MS, respectively. AGC target values were set to 3 × 10^6^ and 20 precursors were included using a normalized collision energy of 27. The lock mass at *m*/*z* = 445.1200 was used as an internal calibrant. Data have been uploaded to ProteomeXchange [41] (PXD027289) and can be accessed via Username: reviewer_pxd027289@ebi.ac.uk/Password: 5DWyH5FJ).

## Figures and Tables

**Figure 1 ijms-23-01469-f001:**
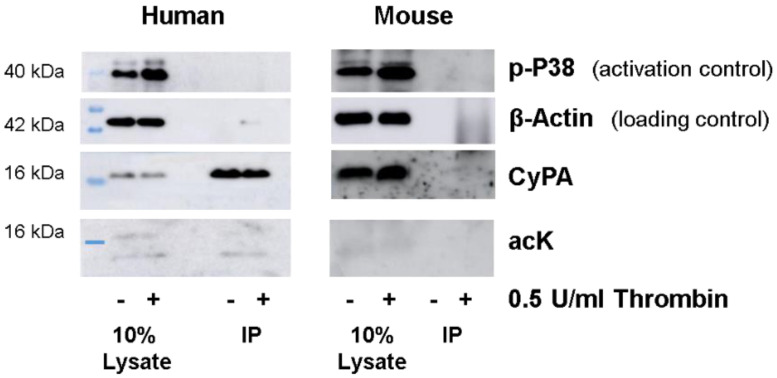
Immunoprecipitation and lysine acetylation of cyclophilin A from human and mouse platelets before and after thrombin stimulation. Human and mouse platelets were stimulated with thrombin. Activation was validated by P38 phosphorylation. β-actin served as loading control. CyPA was detected in human and mouse cell lysates. Only human CyPA was successfully immunoprecipitated (IP) from lysates. Increased lysine acetylation (acK) was not detected.

**Figure 2 ijms-23-01469-f002:**
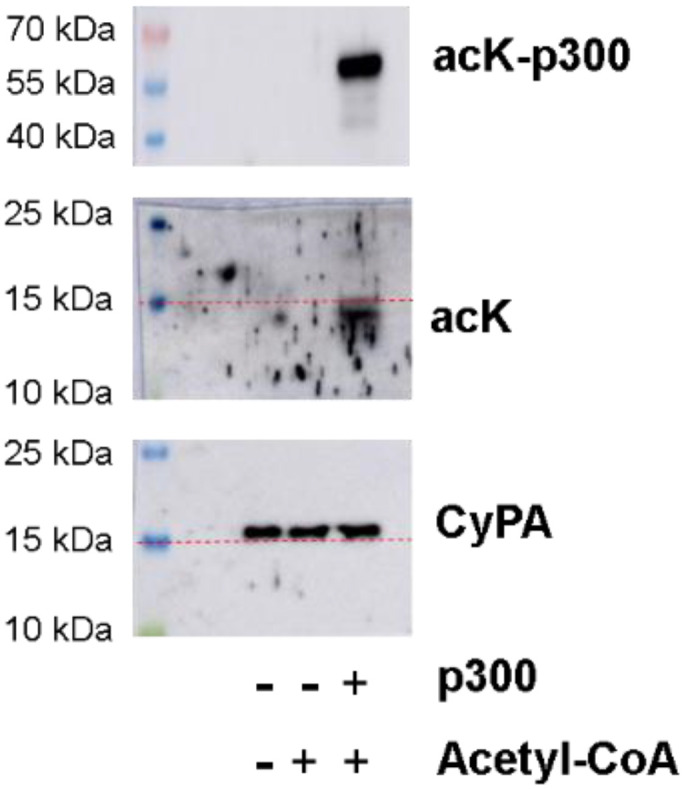
In vitro acetylation of Cyclophilin A by acetyltransferase p300. Recombinant human CyPA was incubated with acetyltransferase p300 in the presence or absence of acetyl-CoA for 30 min. Samples were separated on a 13% acrylamide gel and processed for CyPA and lysine acetylation (acK) Western blots. Mainly auto-acetylation of p300 was observed. The acetylated band at 14 kDa (below red line) was not detected by CyPA antibody (16 kDa, above red line). The experiment was repeated three times with similar results.

**Figure 3 ijms-23-01469-f003:**
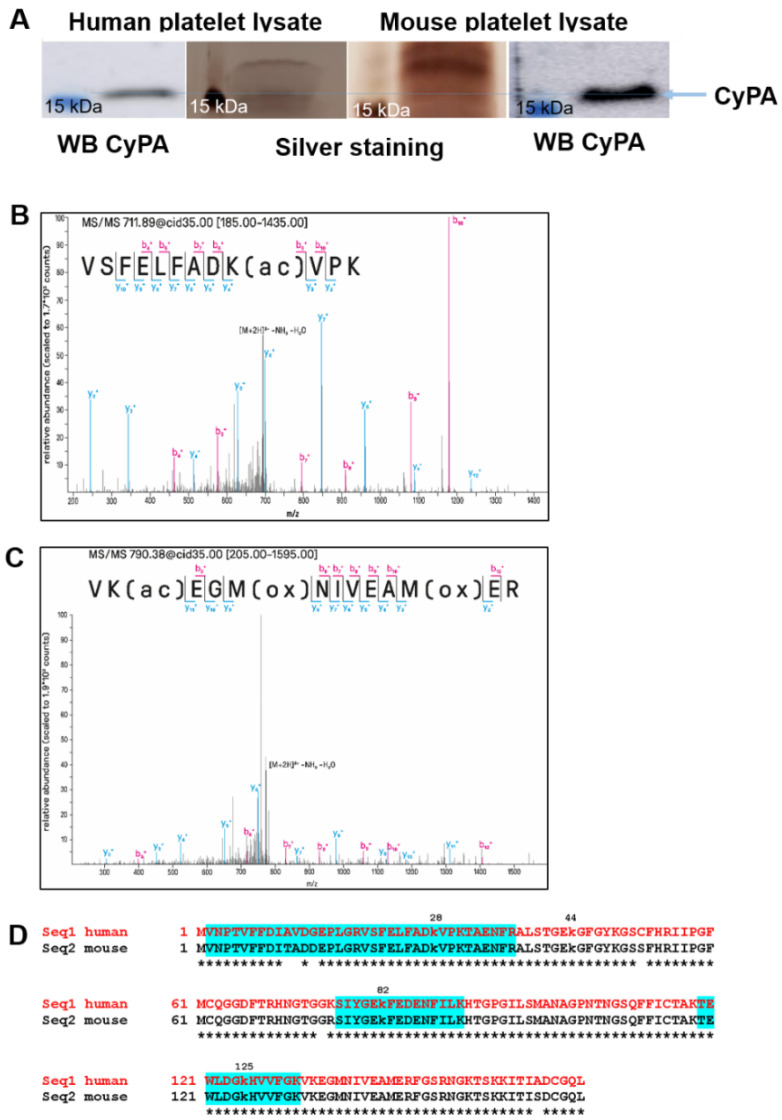
Detection of acetylated lysines in Cyclophilin A. (**A**) cyclophilin A Western blotting and silver staining of human and mouse platelet lysate. Human and mouse platelet lysates were separated on a 13% acrylamide gel, stained with MS-compatible silver nitrate or processed for CyPA Western blotting. After matching, silver-stained potential CyPA bands were cut out for further analyses by mass spectrometry; (**B**) MS/MS spectra of the peptide VSFELFADK(ac)VPK identified with 2 spectra matches and ion scores of 59 and 41 via Mascot database search against UniProt human by low-resolution tandem MS (LTQ Orbitrap Velos Pro) and high-resolution MS, with a mass error of +0.81 and +0.41 ppm; (**C**) MS/MS spectra of the peptide Vk(ac)EGM(ox)NIVEAM(ox)ER, which was likely to be false positive with non-matching precursor isotope-pattern, low ion scores of 35 and 22, and mass errors of −0.07 and −1.95 ppm (mean mass deviation of analysis +1.2 ppm). MS analysis was performed twice with different platelet preparations; (**D**) sequence alignment of human and mouse Cyclophilin A with known acetylated lysines (k). Detected peptides are highlighted in blue.

**Figure 4 ijms-23-01469-f004:**
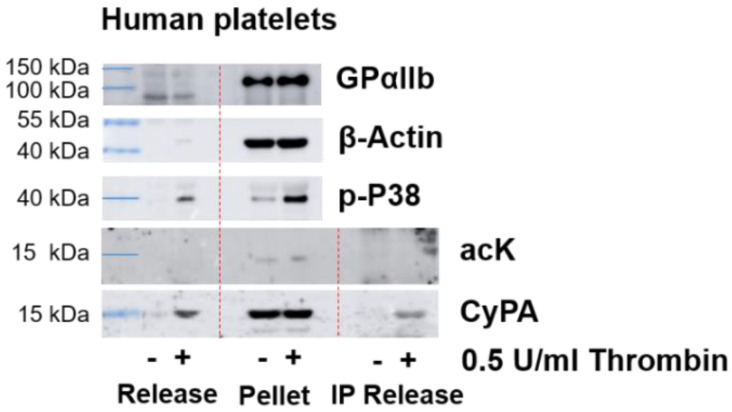
Western blot analysis of released Cyclophilin A in platelets. Platelets were stimulated with thrombin for 5 min. Release of CyPA from activated platelets was analyzed by Western blot, demonstrating CyPA secretion only in activated platelets. No lysine acetylation (acK) of CyPA was observed after stimulation in the platelet pellet or in the immunoprecipitated CyPA (IP) from the releasate. Purity of the released fraction was confirmed by absence of platelet membrane-specific marker GPαIIb. Activation was validated by P38 phosphorylation. β-actin served as loading control.

**Figure 5 ijms-23-01469-f005:**
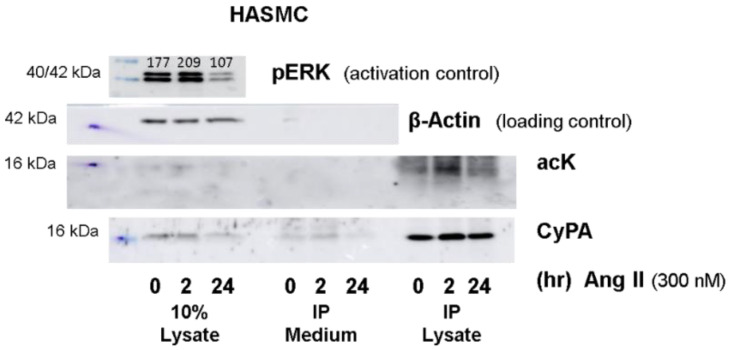
Western blot analysis of released Cyclophilin A in HASMCs. HASMCs were stimulated with 300 nM Ang II for 2 and 24 h (hr). Acetylation of CypA and release into the medium was analyzed by Western blot after immunoprecipitation, demonstrating acetylation of CypA in the lysate, and secretion of CyPA into the medium. Cell activation was validated by ERK phosphorylation; β-actin served as loading control.

## Data Availability

The data presented in this study are available on request from the corresponding author. The data are not publicly available due to privacy.

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
