# Peer review of "Cyclophilin A Is Not Acetylated at Lysine-82 and Lysine-125 in Resting and Stimulated Platelets"

_ijms, 2022, doi:10.3390/ijms23031469_

Round 1
Reviewer 1 Report
In the original research paper entitled „Cyclophilin A is not acetylated at Lysine-82 and Lysine-125 in 2 resting and stimulated platelets” by Dr Rosa et al., the Authors analysed a potential role of acetylation of CyPA protein in blood platelets. Using Western blot and mass spectrometry, the Authors showed no association between the acetylation of CyPA and its release from platelets stimulated with thrombin both for human as well as mice platelets. In my opinion, the study is well designed and the paper is well-written. I strongly support a rationale to publish negative results. I do not have any specific criticisms regarding this study, I have only one remark. Maybe it is worth considering to investigate whether CyPA could be acetylated in K85 and K125 in vitro by acetylsalicylic acid (ASA) or other acetylating agents? It seems to be interesting just by analogy to COX-1 susceptibility to acetylation.
Reviewer 2 Report
The study by Rosa et al identifies that the acetylation of CyPA is not a major protein in platelets and is redundant for its secretion from platelets. While the study is well designed, the quality of the presentation can be improved for better clarity. The manuscript appears to be rather confusing in its present form. Most importantly, the findings of this study rest highly on the western blots, which are of poor quality and may therefore affect the interpretation of the outcomes. While there are always issues with the affinity of the antibodies, it is still advised to improve the quality of the blots even marginally. Some of the comments are.
- The introduction written by the authors is informative to understand the role of CyPA. However, the rationale of the study and overall hypothesis of the manuscript is missing. The entire reason for this study is mentioned rather briefly in a few lines. (Page 1, Line: 74-76). It is therefore required to mention these to bring clarity into the manuscript.
- The blot for the detection of the acetylated form of CyPA is of poor quality (especially in the case of mouse platelets, where the band looks more like a smear). It is therefore advised to provide better quality blots.
- The authors suggest that “no increase in acetylation of CyPA was detected after thrombin stimulation” – This is hard to suggest from the present quality of blots.
- Figure 3 does not add much to the manuscript, and it can be merged with figure 4.
- Figure 4 is numbered wrongly.
- Figure 5: Was the IP performed for acK performed on the releasates? If so, it shows the acetylation of CyPA.
- Isn’t The findings in figure 5 opposite to what was suggested in Figure 1, where it is shown that lysine acetylation (acK) of CyPA exist in resting and activated platelets. Isn’t it surprising that lysine acetylation (acK) of CyPA is absent in both releasate and pellet?
Round 2
Reviewer 2 Report
The authors have addressed all my comments. I have nothing more to add.